# Stakeholder Perceptions of the Challenges to Racehorse Welfare

**DOI:** 10.3390/ani9060363

**Published:** 2019-06-17

**Authors:** Deborah Butler, Mathilde Valenchon, Rachel Annan, Helen R. Whay, Siobhan Mullan

**Affiliations:** School of Veterinary Sciences, University of Bristol, Langford, North Somerset, Bristol BS40 5DU, UK; mathilde.valenchon@bristol.ac.uk (M.V.); rachel.annan@bristol.ac.uk (R.A.); Bec.Whay@bristol.ac.uk (H.R.W.); siobhan.mullan@bristol.ac.uk (S.M.)

**Keywords:** racehorse welfare, staff shortages, horse–human relationship, standards of care, employee relations

## Abstract

**Simple Summary:**

British horseracing industry stakeholders were asked to discuss some of the challenges they perceived as having an effect on the welfare of racehorses in training. A shortage of racing staff was mentioned in six of the nine themes stakeholders identified as having an effect on racehorse welfare. Staff shortages were perceived as having an effect on welfare directly, through standards of care given to racehorses in training, and indirectly, through poor employee relations between racehorse trainers and staff, perceived as affecting attitudes and behaviour which, in turn, can affect the welfare of horses in training and potentially their performance.

**Abstract:**

The purpose of this paper is to highlight some of the key challenges to racehorse welfare as perceived by racing industry stakeholders. The paper draws upon statements and transcripts from 10 focus group discussions with 42 participants who were taking part in a larger study investigating stakeholders’ perceptions of racehorse welfare, which participants recognised as maintaining the physical and mental well-being of a performance animal. Analysis of the 68 statements participants identified as challenges produced nine themes. Among these, 26% (18 statements) of the challenges were health related, whilst 41% (28 statements) focused on the effect staff shortages were having on the racing industry. Staff shortages were perceived as affecting standards of racehorse care and the opportunity to develop a human–horse relationship. Poor employee relations due to a lack of recognition, communication and respect were perceived as having a detrimental effect on employee attitudes, behaviour and staff retention which, in turn, can have a sequential effect on the welfare and health of horses in training. Although the number of challenges produced is small (68), they emphasise the perceptions of stakeholders closely associated with the racing industry.

## 1. Introduction

The British racing industry has been said to have been experiencing an ongoing labour shortage from at least the early 1970s [1,2,3,4], a shortage that has yet to be resolved. The aim of this paper is to highlight some of the effects staff shortages were perceived as having in racing yards and how a shortage of labour and poor employee relations may affect horse husbandry and potentially racehorse welfare.

The role of stable staff as carers is an area that has received relatively little academic attention even though research has shown that animal carers/stock people can have a major impact on the welfare of the animals in their care [5]. These attributes play an important part in working practices that promote positive interactions between humans and, as in this study, horses. Drawing on data collected during focus groups with racing industry stakeholders, the aim of this study is to provide an insight into one of the main challenges racing industry stakeholders identified, that of staff shortages, and the impact this is perceived as having on horse welfare. Maintaining a standard of care in the face of structural changes in the organisation of work in racehorse training yards is of increasing importance to the racehorse welfare debate and is an area that has become visible during discussions with racing industry stakeholders.

At present, there are 550 racing yards training on average 16,221 racehorses per month in Great Britain [6]. The industry is mainly based in rural areas and has three main training centres with communal gallops: Lambourn, Newmarket and Middleham. Training racehorses is a challenge in itself, a precarious profession in that, to use a well-worn epithet, ‘you are only as good as your last winner’. Trainers’ income is very variable and is mainly derived from training fees, owners and, if they are successful, a percentage of any prize money their horses might accrue. If the trainer has a bad season or trains horses of moderate ability, the amount of prize money received may be negligible. Poor results can result in owners moving their horses to another trainer, thus reducing income coming into the business.

Training racehorses is a relatively labour-intensive occupation. In 2018, there were 6734 registered employees, 4428 full-time and 2306 part-time, of which 3493 were male and 3241 were female [7]. Employees in the racing industry have to be registered with the British Horseracing Authority (BHA) by their trainer on the Register of Stable Employee Names [8], a procedure not seen in other sections of the equine industry. Stable staff have a trade organisation, The National Association of Racing Staff (NARS), ‘the trade union for racing staff’ [9] who negotiate with the National Trainers Federation (NTF), ‘the voice of Britain’s racehorse trainers’ [10] over, for instance, pay and conditions, hours of work and other work related policies.

## 2. Staff Shortages and Changes in the Organisation of Work in the Racing Industry

At present, it has been estimated that the racing industry has a shortfall of approximately 500–1000 available stable staff. Whilst steps are being taken to improve domestic recruitment of stable staff, staff shortages may be may be further exacerbated should changes be made to British immigration policy which restrict the number of workers, as the racing industry has for many years employed migrant workers from 23 European Economic Areas (EEA) excluding the United Kingdom as well as non-EEA countries to supplement its shrinking workforce [7].

Staff shortages, however, are not a new phenomenon. In 1974, The Committee of Inquiry into the Manpower of the Racing Industry (CIMRI) was appointed by the Joint Racing Board (JRB) as ‘a result of the growing concern throughout the racing industry about the current and future labour position’ [1]. Although the most severe shortages of staff were found in Newmarket, where the highest proportion of horses were in training, there was a country-wide shortage of labour that was thought to be caused by an inability to attract and retain suitable stable staff. Faced with a shortage of staff, some trainers were beginning to employ girls, something of which the JRB [1] was aware. Women began working as stable staff in the mid-20th century when the racing industry was faced with a shortage of male labour; women’s suitability for this work was couched in terms of their ‘natural love of horses’ [1].

Research by Filby [11] identified other significant changes that were having an effect on the organisation of work in racing yards. These included, for instance, the introduction in the late 1970s of a national minimum wage for racing staff, which meant trainers found themselves having to adapt their labour processes rather quickly once a national minimum wage had to be paid. Indentured apprenticeship, once the only entry route into racing for small, lithe young men with aspirations of being a jockey, was abolished in 1976, thus bringing to an end the superexploitation of indentured apprentices. The once ready supply of cheap labour (indentured apprentices) stopped (see Butler [12] for a more detailed explanation of indentured apprenticeship). Other factors played their part: a declining workforce lured by the opportunities available of alternative better-paid employment, an increase in the size and weight of the population and an increase in the number of horses in training [6,11]. In 1975, there were 11,491 horses in training; this has now risen to over 16,000 in 2018 [6,13] which, coupled with the introduction of Sunday racing in the 1990s [14], an increase in the fixture list from 1132 meetings in 2000 to 1508 in 2018 [6,13,15], a greater regulation in the hours stable staff can work and overtime payment, [9] has meant staff and trainers are constantly under pressure.

All of these changes meant that the historic practice and custom of ‘doing your two [horses]’ is no longer used as a standard measurement of work as set out in the Memorandum of Agreement between the National Trainers Federation and The National Association of Racing Staff [9]. The practice involved one member of staff having responsibility for their ‘own two horses’ which they would muck out in the morning, ride them on exercise and feed them at lunch time (12:30–1:00 p.m.). Two or three ‘lots’ (exercise routines) would be the norm. In some yards, staff were not to go back into the yard until ‘evening stables’, as it was viewed as the time horses would be able to rest during the day. Evening stables typically started at 3:30 p.m. and finished at 5:30–6:00 p.m., sometimes later. Staff would skip out their two, giving their horses a ‘dressing over’, that is, a thorough groom, possibly strapping them with a wisp made from plaited hay, then leave them tied up until they were told to let the horses down, that is, untie them. The head ‘lad’ would, during this time, check each horse over, asking the lad if he or she had noticed or sensed if anything was amiss. The trainer would often ‘look round’, typically starting at 5:15 p.m. when he or she would enter every box, check the horse over whilst it was being held by the lad (see [12] for more detail). Once the procedure had finished, the head lad could then feed the evening feed with each lad feeding their two horses. The weekly working week was set as 48 h per seven days, although most racing staff would have typically worked over that with no overtime paid.

The title of ‘stable lad’ (male and female) has now been changed to ‘racing grooms’ [16] and has given way to a different occupational hierarchy [1,2,11,12]. This has involved a disaggregation of work roles, a polarisation of skills and knowledge and the creation of different roles within the yard. For example, yards will typically employ yard staff who will only muck out and tidy the yard, ‘rider outers’, or ‘work riders’, who only ride out in the mornings, paid by the number of exercise lots they ride, together with full-time staff who will carry out a more traditional composite role within the yard and will have five or more horses assigned to them, although not always to ride out. Evening stables will typically last no more than two hours and involve staff skipping out five horses, ‘setting them fair’, that is, brushing off any sweat marks, picking out feet and removing and replacing horses’ rugs. If staff are away racing or, as now happens in some yards, a certain percentage of staff get an afternoon off if they have worked a weekend, the remaining staff will ‘work round’, that is, carry out the basic husbandry routines such as skipping out, putting forage in, checking water and straitening rugs. The head lad or assistant trainer will move around the yard and check each horse’s legs, check they have eaten up and generally appear healthy and settled.

There are very few biographies describing what life as a racing groom, a stable lad, is like, who perform similar roles to stockpeople in the livestock industry. It is therefore reasonable to assume that relationships similar to those reported in a number of livestock industries may exist in recreational and working horse populations. Whilst there are differences in the degree of human interaction that exists with management and husbandry practices both within livestock industries as well as between horse and livestock industries, the frequency and quality of interactions and the context in which they occur will determine the quality of the relationship [17].

Hemsworth et al. [5] discussed the topic of ‘stockmanship’ in farm animal welfare monitoring schemes, highlighting how, as a topic, it has received relatively little attention even though research has shown that stockpeople have a major impact on the welfare of their stock. As Seabrook [18] found, the welfare of farm animals is dependent upon the actions of stockpeople, who regularly handle, observe and monitor the animals in their charge. Research to date has focused upon the more obvious aspects of stockmanship: how the animals are handled and how they become fearful of people. However, stockmanship involves much more, and the relationships that can develop between people and animals can be quite subtle [19,20,21].

The use of horses, in principle, differs little from the use of other animals for food, transport or entertainment [22,23,24]. The racehorse is no exception. It is the central player in a complex relationship that revolves around, at a macrolevel, the betting industry and Thoroughbred horse breeding, and at a microlevel, the racehorse trainer, the racehorse owner, the jockey and the stable staff. All parties are reliant on the racehorse to provide their leisure, employment and financial security. Given their role within the racing field, the racehorse could be defined as a production animal, an ‘ambiguous commodity’ [25]. The racehorse as a specific breed, the Thoroughbred, was and still is subject to asymmetries of power where their genealogy, their working and reproductive life (if they have one) and ultimately their death is dominated by a political ecology of human dominance and exploitation in the same way livestock can be.

This study identifies the effect staff shortages were perceived as having on racehorse welfare, firstly, on standards of care and, secondly, on employee relations in some racing yards. It highlights the challenges in maintaining positive contacts between horse and human despite the trend in racing to increase the number of horses each staff member looks after, as has been seen to be the case in agriculture [26].

## 3. Materials and Methods

### 3.1. Participant Recruitment and Response

This study drew upon statements collected from an exercise that formed part of 10 focus group discussions with 42 participants who were taking part in a larger study investigating stakeholders’ perceptions of racehorse welfare (see Butler et al. [27] for more details of the study). Participants were recruited to reflect the main stakeholder groups within the industry, which included equine veterinary surgeons; racehorse trainers; assistant trainers; stable staff; owners; human and animal nongovernmental organisations, for instance, Racing Welfare and World Horse Welfare, respectively; paraprofessionals such as equine physiotherapists; BHA veterinary officers and BHA racing yard inspectors. Potential participants were contacted through a variety of methods. One of the authors (D.B.) had worked and is still closely involved in the racing industry and was able to draw upon personal contacts for stable staff and ancillary racing stakeholder groups, who in turn asked their friends if they would like to attend the focus groups. Such an approach could best be described as a snowballing technique [28]. Other participants were contacted through their place of work via email. The focus groups were posted on Facebook by Racing Welfare, the charity which supports the workforce of British racing. Posters were also displayed by Racing Welfare in Newmarket and Middleham. Holding focus groups in the main racehorse training areas and centrally in Britain ensured that, theoretically, attitudes and perceptions of welfare could be gathered from a range of racing industry participants. To facilitate a more open discussion, two separate focus groups were held in each training centre, one for trainers and one for stable staff and ancillary racing personnel who had worked in racing. These were held in either local pubs or hotels. In addition, a combined focus group was held in the Midlands, as there are a plethora of racing yards in these areas; one in London for representatives from equine charities and BHA veterinary officers; another at a University Veterinary School for members of their Equine Veterinary Hospital who work with racehorse trainers and as specialists in the Equine Hospital; and the tenth was in the Cotswolds for BHA stable inspectors, BHA veterinary officers who work at the racecourse and paraprofessionals who work with racehorses. Racing’s stakeholders are located across the length and breadth of Britain; focus groups were thus a more resource-efficient way of reaching a wider spread of participants when compared to carrying out face-to-face interviews.

### 3.2. Structure of Focus Group Discussion

Participants were given a participant information sheet and signed a consent form which informed them of their right to withdraw from the focus group. The study and consent process had been given ethical approval by the University of Bristol Faculty of Health Sciences Ethics Committee (R113851-101). Each focus group was run by at least two members of the research team, including a trained facilitator (S.M.) for the first eight.

Participants worked together on three exercises. The first scenario-based exercise asked participants to imagine themselves as a racehorse in training. As a racehorse, participants were asked to identify within their group what important elements would form the minimum welfare standards they might be kept under, and conversely, what would contribute to the ‘best life’ they, as imaginary racehorses, might experience, where money and other factors were no object. These were written down on sticky notes by participants as they discussed the scenarios together and placed on a flip chart paper divided into two sections marked ‘minimum welfare standards’ (MWS) and ‘best life’ (BL).

This study drew upon data collected from the second exercise, in which participants were asked to individually identify the three main welfare challenges racehorses faced. The challenges were written on sticky notes and put up onto a large flip chart to facilitate a group discussion.

### 3.3. Data Analysis

Using thematic analysis, nine themes were identified from the 68 statements that were provided and ranked according to number of statements (Table 1).

Out of nine themes, health was perceived as the main challenge, although 11 statements relating to staff shortages were also associated with five other themes. These were training, exercise and recovery, daily routine and monitoring, physical comfort/living environment, turnout/social contact and feeding. In total, 41% (28 statements) of the statements were staff related.

Further thematic analysis by D.B. of the six themes which contained staff-related challenges identified two strands, standards of care and employee relations, areas which were perceived as impacting upon racehorses in training and thus on their welfare.

Once the two strands had been identified, audio recordings from the 10 focus groups were transcribed verbatim and analysed by D.B. Transcripts were read through and first coded manually to the nine themes initially identified. Sections of the transcripts assigned to each of the six themes were then recoded manually to the two identified strands of standards of care and employee relations.

## 4. Results

In terms of health, veterinary surgeons and trainers identified similar challenges such as ‘soundness’ and ‘maintaining good health’ and the avoidance of ‘disease, flu/herpes and low-grade bacterial infection’. The veterinary surgeons outlined how, for a horse in training, factors such as ‘veterinary aspects (pain, disease) [were] more important welfare issues than management aspects’ where ‘on-going health issues such as stomach ulcers and repetitive injuries cause lack of performance’.

The results shown below drew on some of the 68 statements participants perceived as challenges to racehorses in training together with an analysis of the transcripts from each focus group to illustrate participants’ perceptions of staff shortages and the impact they see these having on welfare.

### 4.1. Standards of Care

Many of the factors participants identified were related to standards of care and how, if levels of horse husbandry are not maintained, horse welfare may be affected. Statements such as ‘the staff-to-horse ratio’ and ‘lack of staff to look after horses in the yard when others [are] off racing’. However, as participants were aware, some trainers cannot afford to employ more staff to care for the horses they have in training:
“In an ideal world you’d have one lad to every three, you now have one lad to every four or five. I think that’s a reasonable number. In a yard that’s struggling financially, your minimum is probably one lad to every eight (female racing staff).”

Trainers and staff alike were outlined as ‘not knowing their horses’, with participants highlighting ‘how the lack of time spent out on exercise’ was in some instances a challenge. Trainers acknowledged they had to adjust their general organisation of work as staff are perceived as not being available:
“We’ve had to adjust our work routines to accommodate what’s really happening out there and there’s a massive staff shortage. You’ve got to have a way of adjusting your routines to use staff differently (racehorse trainer).”

As another trainer explained:
“You do the warm up and cool down on an electric walker while your staff are out on another lot (racehorse trainer).”

Participants were acutely aware how a ‘lack of knowledgeable experienced staff’ and ‘few experienced staff’ can have an impact on standards of care and welfare and was something that was identified frequently:
“I know there’s nothing we can do about it, and that [staff experience] it’s something in decline [and] that becomes a welfare issue. Things get missed, I’ve been in big yards where we’ve had enough staff, we’ve not had the one-to-three ratio, but we’ve had enough and things get missed. Someone doesn’t pick a horse’s feet out and then the next day it comes out and it’s got an infection that’s gone into its white line and it’s lame (female participant).”

Participants felt that the lack of staff meant corners were bound to be cut:
“If you’re doing a line of six horses in the evening where you’re running through them as fast as you can, because there’s a million and one things to do in the hour, I feel like that’s having a negative effect on the horses indirectly (female racing staff).”

As one participant outlined in a statement, there are ‘poor standards of care around horse husbandry when the stable routine is compromised through lack of staff’. Other statements such as, ‘standards of facilities and standard of stables’, ‘cleanliness of yard’ and ‘the level of health and welfare in a yard being affected by yard hygiene and the incidence of low-grade bacterial infections’ were areas affected by a shortage of time available caused by a lack of staff to make sure routines were completed properly. Comments such as:
“The yard and different areas like the feed room don’t get swept, cleaned out properly and the same with mucking out. Horses cope with it, lying in shit, but I can’t see as it’s good for them (male racing staff).”

It was thought that, if a situation such as this were to be avoided, the experience of staff employed was seen as vital in maintaining welfare:
“And the mix [of staff] as well I guess, in that you could have, you know 10 horses, three knowledgeable staff and two kids who are working in the evening after school, then that’s great but if you’ve got five kids that are all working in the evening after school and nobody else, that’s not good (male racing staff).”

The lack of experience was perceived as having a direct impact on horse welfare:
“But if you take it from a welfare point of view the issue is labour, there is not enough labour full stop in terms of skilled labour (male racing staff).”

Two written statements focused on how ‘the lack of contact time with horses’ and how ‘the lack of contact time between horse and handler affected their ability to build a horse–human relationship’. These were areas participants identified as potentially limiting the horse–human relationship that many stable staff see as an important and integral part of their daily routine. As one participant illustrated:
“I’m a very strong believer in horses getting a really big kick out of their person that’s in their life every day. There’s proof of that in the way your horses react to you. I had a colt at XXXXX and he’d call to me every time he heard me calling in the yard. They get a buzz out of that and I feel like the staff retention situation is affecting the horses in that way as they [horses] don’t get one-to-one (female racing staff).”

The horse-to-staff ratio was perceived as an area that could compromise welfare and participants were aware how staff shortages can affect the horse–human relationship:
“We put here the staff ratio possibly 3 horse to 1 staff max. So that you really have time to know your horse and develop a relationship and understand their likes, dislikes (female racing staff).”

### 4.2. Employee Relations

In terms of employee relations, two written statements thought ‘bad and incompetent trainers’ could be detrimental to welfare, as a former, experienced member of staff expressed:
“I think there’s almost a hard-nosed attitude towards staff, and if you don’t look after your staff they can’t look after the horses and they’re not going to want to do it for you, go that extra mile. We all know (female racing staff).”

Communication and trainers not listening was a bone of contention, highlighted four times, for instance, ‘trainers not listening when they are told a horse is lame’ as well as ‘poor employment relations’ where ‘yards should be investigated when there is a high number of staff down as leaving’. As one member of staff explained:
“I stopped saying something to the trainer because I can protect the horse more by shutting up. I go steady and look after him/her a little bit (male racing staff).”

## 5. Discussion

The aim of this study was to highlight one of the key challenges to racehorse welfare as perceived by racing industry stakeholders, that of staff shortages and the organisation of work and its perceived effect on horse husbandry and thus on racehorse welfare. Nevertheless, what must be acknowledged is that the perceptions and practical reasoning of the stakeholders are based on their knowledge and experience, their taken-for-granted assumptions which help to define the racing field even if their assumptions may be contingent and arbitrary.

Within the larger study, participants perceived understanding of welfare focused around maintaining the physical and mental well-being of a horse living the ‘best life’ in training, which was strongly associated with racing performance [29]. Given stakeholders’ understanding of welfare in this way, it is not surprising that health, that is keeping an animal free from illness and disease, should be identified as a challenge to welfare.

Little research has ever been carried out on racehorse training as a business, and the nature of racehorse training means it is labour intensive with wages of stable staff representing the largest expenditure item on a trainer’s balance sheet [30]. As has been highlighted earlier in the paper, there is a shortage of staff entering and being retained within the racing industry [4]. Wages too, for experienced staff, are also higher [9]. Trainers may not also be able to afford to have a horse-to-staff ratio of two to three horses; this is now typically four to five horses to one member of staff with a reliance on part-time staff [30].

As was highlighted by participants, it was thought that ‘standards of facilities and stables were being compromised by lack of staff’, thus allowing for the potential spread of infection and disease. An increase in the number of horses to care for means that standards of care in a yard do, as was highlighted, slip, and one of the key aspects of disease control is decreasing exposure to pathogens, which at the most basic level includes the provision of a clean living environment for the horse. Many disease-causing pathogens can be transmitted indirectly through sharing of contaminated fomites [31], which could include horse equipment, clothing or bedding [32]. A lower number of staff means there are more horses per person to muck out and care for, which means less time can be spent in each box, especially when the morning routine must be completed in time to ride out.

As participants highlighted, when a yard has a shortage of staff, it means that less time could be given to each horse, thereby limiting the opportunity to build up a horse–human rapport, seen as an integral part of the horse–human relationship [33,34,35,36]. As Waiblinger et al. outlined [21], the human–animal relationship can be defined as the degree of relatedness between or distance between the animal and the human, that is, the mutual perception which develops and expresses itself in their mutual behaviour and experience with a particular human [36], as one participant illustrated, when ‘her colt recognised her’ as he heard her voice. She will have looked after him on a daily basis, knowing his habits and routine quite often from the first day the colt entered training. A positive human–animal relationship makes it possible for the stockperson, in this case, stable staff, to determine earlier deviations in the animal’s behaviour, which might express the first disease symptoms [37,38]. A good relationship with the animal can therefore be associated with a better health status [39].

If a trainer, whether through affordability or yard reputation as a poor employer, is short of staff, the development of an established horse–human relationship requiring mutual recognition will be harder to establish. This in turn means that the feedback effect created through a close positive human–horse relationship cannot be established, which in turn can lead to an increase in fear and suspicion when, for instance, a horse is being handled, resulting in increased levels of anxiety for horse and handler [40,41]. Whilst some animals, such as the participant’s colt, may also have generalised his experiences, this does not mean he was not able to differentiate between her and other members of staff [41].

If the human–animal relationship can be conceptualised in terms of inter-individual relationships [42], where the quality and frequency of interactions between the two individuals as well as the context in which they occur, can determine the quality of the relationship, the low human to high horse ratio identified by participants in some yards where trainers are short of staff may make building up a human–animal relationship difficult, as seen in with other livestock [43]. Studies into the behavioural effects of interactions between humans and horses has been a topic of investigation in recent years [17,35,36]. Many have speculated that a person’s attitude and confidence level will affect the behavioural reactions of the horse [44,45,46]. In a similar vein, Hausberger et al. [3] highlighted that whilst there is no prescriptive method of being able to adapt and handle horses with different temperaments [39], only well-trained observational skills allied with advanced knowledge of horse behaviour can mean horses can be handled safely in terms of horse–human interactions [39,47]. If this is the case, the association participants made between the lack of experienced staff and potential detrimental effects on welfare may be substantiated. Participants were in agreement that a lack of knowledgeable experienced staff was a challenge and one which could have an impact on welfare and performance. Whilst literature on the role of stable staff in maintaining and promoting horse welfare is extremely limited both from a social and animal science perspective, as Hemsworth et al. [17] identified with recreational horse welfare, there are possible relationships between horse owner attributes and horse welfare outcomes. There are a small number of studies on recreational horse owners where one of the key factors in horse–human relationships and its link to the health and welfare of recreational horses may be owners’ attitudes and their performance of husbandry and management practices [17,38].

Employee relations, that is, a company’s efforts to manage relationships between employers and employees [48,49], were said to be poor by some participants, which can affect the retention of staff and thus the number of employees available in a yard. Working closely with any animals means staff will often make an extra effort, ‘go that extra mile’ as the participant highlighted. When being conscientious and empathetic is routinely not recognised and staff are constantly criticised, as is implied within the quote, attitudes and behaviour towards the horse can be affected [19]. For instance, one participant discussed the way in which staff ‘protect’ the horse they are riding by ‘shutting up’. What is being referred to is the fact that the rider felt the trainer was not listening to the feedback the rider was giving him about how the horse did not seem to be moving very well on exercise. The reference to ‘shutting up’ means that when the trainer asked the rider again when the horse was exercised how he/she was moving, the rider will have just said ‘fine, moving okay’, although in the rider’s opinion there was no improvement. As the trainer will think the horse is fine, the horse’s exercise will reflect this. However, the rider will ignore the instructions from the trainer by going at a slower pace than the trainer has instructed the rider to do, thus ‘protecting’ the horse when the horse did not seem to be moving correctly. This scenario has implications both for the welfare of the horse and for employee relations. The goal of a racehorse trainer is to produce winners; having riders tell him or her a horse is lame is not what he or she wants to hear. Nevertheless, for staff members, their experience and knowledge are ignored and their concern for the well-being of the horse is ignored, creating a barrier to two-way communication and job satisfaction.

## Ethical Approval

The study and consent process have been given ethical approval by the University of Bristol Faculty of Health Sciences Ethics Committee (R113851-101). Only adults (>18 years) were involved; all participants took part on a voluntary basis and could withdraw at any time. Information regarding the purpose, intent, motivation, funding body, potential use of data and methods of data collection were provided to participants prior to the beginning of the focus groups. All data were collected anonymously and it was not possible to identify participants in the raw research data.

## Figures and Tables

**Table 1 animals-09-00363-t001:** Number of statements per theme and associated frequency ranking.

Associated Freq. Ranking	Themes	No. of Statements	Statements/Theme (%)
1	Health	18	26%
2	Staff management and education	17	25%
3	Training/exercise and recovery	9	13%
4	Daily routine and monitoring	8	12%
5	Physical comfort/living environment	6	9%
6	Policy and procedures	3	4%
7	Turnout and social contact	3	4%
8	Owner/breeders	2	3%
9	Feeding	2	3%
Total		68	100%

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
