# Peer review of "Stakeholder Perceptions of the Challenges to Racehorse Welfare"

_animals, 2019, doi:10.3390/ani9060363_

Round 1
Reviewer 1 Report
An interesting study of the perceptions of stakeholders in the horseracing industry, which potentially makes an important contribution to scholarship in human-equine relationships. It is well written and properly referenced. I do, however, have some criticisms of it; I suggest that it needs some revision to ensure that the arguments flow.
First, the focus of the paper is not entirely clear. Although the abstract says it is about stakeholders' perceptions of welfare, much of the paper focuses on organisation of work in the racing industry. Either would be interesting, but it isn't obvious how work organisation impacts perceptions of welfare. It would help to have more about how stable staff work with horses in the industry, to help tease apart these threads.
I found the stated aims confusing. For example, the abstract refers to an aim focused on welfare, yet on p 6 (discussion), it says that the aim is to analyse the impact of staff shortages. Which is it? If both, then that needs to be spelled out. On the whole, I felt that the paper did not say very much about welfare itself. This may be in part because there was no clear definition. It is interesting that stakeholders talked about welfare in terms of health. This is undoubtedly important, but also might be said to be a limited, even old-fashioned, focus; "welfare" is much more than physical health, however important. The referenced studies in agricultural animals, for instance, include studies of stockperson behaviour, and effects on animals' behaviour, not only on outcomes of physical health. Some discussion of why "health" seems to be the primary concern might be interesting; what might they mean by that?
Second, and relatedly, although the paper cites another part of a larger study, there needs to be much more information given here. I wanted to know how participants were selected, what they were asked to talk about. More methodological information needed!
I would also like to have seen a little more nuanced analysis of responses. For example, on p. 6, the speaker implies that he could 'protect' the horse by 'shutting up'. This really needs a lot more unpicking, both in terms of providing welfare, and in terms of work structures.
More specific points:
p. 6: I'm not sure that this study has said anything about impacts of staff shortages (see general point above)
p. 7: again, following on from general points, line 288 makes reference to studies of impact of stockpeople on animal behaviour, but this has not really been discussed with reference to horses.
Further down, around line 330, there is a reference to "poor riding". But this is not defined, and it isn't clear how it relates to the present study.
Author Response
Dear Reviewer
Thank you very much for your comments. My responses are in red.
First, the focus of the paper is not entirely clear. Although the abstract says it is about stakeholders' perceptions of welfare, much of the paper focuses on organisation of work in the racing industry. Either would be interesting, but it isn't obvious how work organisation impacts perceptions of welfare.
I have rewritten the abstract to hopefully reflect the aims of the paper and change the emphasis as it originally written, on health.
It would help to have more about how stable staff work with horses in the industry, to help tease apart these threads.
line 97 -127. I have added a brief description of how routines have changed in the industry and the typical routines now.
ave included a brief description of how
I found the stated aims confusing. For example, the abstract refers to an aim focused on welfare, yet on p 6 (discussion), it says that the aim is to analyse the impact of staff shortages. Which is it? If both, then that needs to be spelled out. On the whole, I felt that the paper did not say very much about welfare itself. This may be in part because there was no clear definition. It is interesting that stakeholders talked about welfare in terms of health. This is undoubtedly important, but also might be said to be a limited, even old-fashioned, focus; "welfare" is much more than physical health, however important. The referenced studies in agricultural animals, for instance, include studies of stockperson behaviour, and effects on animals' behaviour, not only on outcomes of physical health. Some discussion of why "health" seems to be the primary concern might be interesting; what might they mean by that?
I have emphasised the effect staff shortages are perceived as having and changed heading 2, line 66 to make the point more explicit. I have outlined how participants perceived welfare in the abstract and then on p8, line 320-324. I have removed some of the references to agricultural animals and replaced them with references on horse-human relationships and behaviour.
Second, and relatedly, although the paper cites another part of a larger study, there needs to be much more information given here. I wanted to know how participants were selected, what they were asked to talk about. More methodological information needed!
Yes! Sorry about that. I was worried about the fact I may be repeating material twice hence why I left some of the methodology out. I have added much more information to section 3, material and methods and to the section 4,results. I have removed some of the quotes on trainers as they were not really relevant.
I would also like to have seen a little more nuanced analysis of responses. For example, on p. 6, the speaker implies that he could 'protect' the horse by 'shutting up'. This really needs a lot more unpicking, both in terms of providing welfare, and in terms of work structures.
I have explained what the speaker was implying, line 396-405.
Specific points
p. 6: I'm not sure that this study has said anything about impacts of staff shortages (see general point above).
p6. (now p.8) I have removed the line
p. 7: again, following on from general points, line 288 makes reference to studies of impact of stockpeople on animal behaviour, but this has not really been discussed with reference to horses.
I have included a discussion on horses, removed the references to agricultural animals and increased the references relating to horses. line 386-401
Further down, around line 330, there is a reference to "poor riding". But this is not defined, and it isn't clear how it relates to the present study.
I have removed the quote and reference to poor riding. It does have relevance but it would take a considerable amount of unpicking to make its relevance explicit and clear. It also reflects the speaker's frustration too, that no-one actually appears to be concerned about standards of riding. As long as horses are exercised that's all that matters. This however was not made explicit in the focus group or in the transcripts, more from personal conversations with those working in the industry
Reviewer 2 Report
22 Sentence a little confused in its direction
41 Not sure I would describe as relatively small - in what context?
53 BHA - needed in full then abbreviated
61 Is this relevant re the mainly EU origination of staff?
93 Commonly called work riders - I have never heard of the term rider outers? I have been involved with the industry for many years.
95 Not sure of this - 1987? Not my experience in the industry
105 Check this statement
108 Disagree - there have been quite a few recent publications (both references used to support this statement are from 1999...) - with the advent of equitation science there are paper published from 2005 onwards
124 Also a moderate and active Arab/Arab x TB racing community in the UK
148 68 statements is quite a low number to extrapolate trends from
248 Reference 30 - no date so unclear of age of content
253 Reference 32 - 2007? I would like to see a more up to date source used for this point
273 Reference 36/37 - maybe more appropriate with two references regarding horse-human relationships rather than the species used (see later point 283)
275 There is research for this for horses being able to recognise handler/caregivers
277 Unclear evidence - reference equine research
283 References 43-47: these are not really relevant any more (one as old as 1970s). Only 48 is recent, there are many more available
325 Spelling
328 Disagree - now a fair few papers published on this subject across varied equine disciplines
334 Makes no sense - maybe talk about balance and novice riders as this seems appropriate here. I would say based on evidence that a novice rider is probably the cause of great harm to welfare and not a plus as suggested.
Author Response
Thank you for your comments. I have added my responses in red.
22 Sentence a little confused in its direction
I have rewritten the abstract and the line 22 now reads differently.
41 Not sure I would describe as relatively small - in what context?
I have removed 'relatively small' as to anyone involved with the racing industry is does not appear that small. It was a term that was used to describe the racing industry in 2004 and was made by an independent committee member. who worked in the manufactoring industry.
53 BHA - needed in full then abbreviated
Added
61 Is this relevant re the mainly EU origination of staff?
I would argue, yes. I have made reference to the numbers of EEA and non-EEA countries who work in racing who were employed to supplement shrinking numbers of domestic labour. Many yards could not operate without migrant labour.
93 Commonly called work riders - I have never heard of the term rider outers? I have been involved with the industry for many years.
Perhaps its regional?. Where I have worked and have also ridden out 'rider outers' was the term commonly used for casual staff who rode out. In the occupational hierarchy work riders were higher up the ladder, ex-jockeys, or jockeys brought in to specifically ride work. The term however is used more now to refer to part-time members of staff who ride out only which I have referenced in the manuscript, line 118, p.3.
95 Not sure of this - 1987? Not my experience in the industry
I have removed the line regarding de-skilling, work intensification and the increasing use of migrant labour as I have now added in a paragraph on changing working routines. In my experience of working for many years in the industry was employing migrant labour and treating them very badly, as they were young people, enrolled onto Youth Training Schemes who were used as cheap labour for a year, then let go with few tranferable skills. Filby's (1987) paper and his original thesis highlights how changes were starting to creep into the industry.
105 Check this statement
I have taken out the quote from Newby and the sentence.
108 Disagree - there have been quite a few recent publications (both references used to support this statement are from 1999...) - with the advent of equitation science there are paper published from 2005 onwards
Point taken and the text removed.
124 Also a moderate and active Arab/Arab x TB racing community in the UK
Yes although I have only included Thoroughbreds within the manuscript. I thought making reference to another type of racing may confuse readers who have little knowledge of racing.
148 68 statements is quite a low number to extrapolate trends from
I have acknowledged this in a sentence in the abstract.
248 Reference 30 - no date so unclear of age of content
It was 2017.
253 Reference 32 - 2007? I would like to see a more up to date source used for this point
Reference 32. As a point of interest there is a very good piece in the Kingsley Klarion, Mark Johnston's racing magazine, although he doesn't mention wages. He mentions many other things though!
273 Reference 36/37 - maybe more appropriate with two references regarding horse-human relationships rather than the species used (see later point 283)
Yes, I have removed the two references and added in new references, 37, 38 plus 35, 36.
275 There is research for this for horses being able to recognise handler/caregivers
See ref 43
277 Unclear evidence - reference equine research
Added equine references (see below)
283 References 43-47: these are not really relevant any more (one as old as 1970s). Only 48 is recent, there are many more available.
Removed original references and added in more recent and relevant ones,refs 37-46
325 Spelling
Corrected. Paragraph removed.
328 Disagree - now a fair few papers published on this subject across varied equine disciplines
I have removed the paragraph
334 Makes no sense - maybe talk about balance and novice riders as this seems appropriate here. I would say based on evidence that a novice rider is probably the cause of great harm to welfare and not a plus as suggested.
I have removed the paragraph. I may have to disagree with the effect of a novice rider in some cases especially when older experienced 'lads' have ridden the same horse and feel aggrieved thay are riding a 'kids horse'.
Round 2
Reviewer 1 Report
The paper is very much improved, and is now publishable. There are, however, several minor spelling/punctuation issues (duplication and/or missing small words), which can be picked up at copy editing. I suggest the authors go through it first, however.
Reviewer 2 Report
I feel the questions have been answered fully by the authors, however the scientific rigour is still a little basic. The issues surrounding knowledge of equitation science and horse-human relationship have been answered.